# Spin-$\frac{1}{2}$ kagome Heisenberg antiferromagnet with strong breathing anisotropy

Saeed S. Jahromi[1,2*], Román Orús[1,3], Didier Poilblanc[4], Frédéric Mila[2]

**1** Donostia International Physics Center, Paseo Manuel de Lardizabal 4, E-20018 San Sebastián, Spain
**2** Institute of Physics, École Polytechnique Fédérale de Lausanne (EPFL), CH-1015 Lausanne, Switzerland
**3** Ikerbasque Foundation for Science, Maria Diaz de Haro 3, E-48013 Bilbao, Spain
**4** Laboratoire de Physique Théorique, IRSAMC, Université de Toulouse, CNRS, UPS, France
* saeed.jahromi@dipc.org

October 29, 2021

## Abstract

We study the zero-temperature phase diagram of the spin-$\frac{1}{2}$ Heisenberg model with breathing anisotropy (i.e., with different coupling strength on the upward and downward triangles) on the kagome lattice. Our study relies on large scale tensor network simulations based on infinite projected entangled-pair state and infinite projected entangled-simplex state methods adapted to the kagome lattice. Our energy analysis suggests that the U(1) algebraic quantum spin-liquid (QSL) ground-state of the isotropic Heisenberg model is stable up to very large breathing anisotropy until it breaks down to a critical lattice-nematic phase that breaks rotational symmetry in real space through a first-order quantum phase transition. Our results also provide further insight into the recent experiment on vanadium oxyfluoride compounds which has been shown to be relevant platforms for realizing QSL in the presence of breathing anisotropy.

# 1 Introduction

Quantum spin-liquids [1] are exotic phases of matter with highly entangled ground-states and fractionalized excitations which fall beyond the Ginzburg-Landau paradigm [2–4]. Due to lack of local magnetic ordering at zero temperature, QSL are very hard to come by experimentally [1]. However during the past years, there has been tremendous progress at the theoretical level towards a better understanding of these phases, ranging from the QSL ground-states of topological quantum codes [5–9], which are platforms for fault-tolerant quantum computation [5, 6], to highly frustrated quantum spin systems such as Heisenberg antiferromagnets on lattices with triangular geometries [1, 10–13].

The spin-$\frac{1}{2}$ Heisenberg model on the kagome lattice (AFHK) [11, 14–26] is one of the potential candidates for realizing a QSL ground-state in two dimensions. There has been a lot of studies during the past decades to pinpoint the true nature of the AFHK ground-state, and recent studies are in favor of a QSL ground-state [23,24,27–29]. However, the gapped or gapless nature of the ground-state is still under debate. While early density matrix renormalization group (DMRG) results predicted a $\mathbb{Z}_2$ gapped spin-liquid [11,25,26], variational Monte Carlo (VMC) calculations based on Gutzwiller projected fermionic wave functions favoured the existence of a U(1) gapless spin-liquid with algebraic decay of correlations [23,24,30,31]. These results are also supported by current state-of-the-art tensor network (TN) methods [27,28,32] as well as by new DMRG results unveiling signatures of Dirac cones in the transfer matrix spectrum of the ground-state [29].

The AFHK is also of experimental relevance. Recent experiments suggested candidate materials, such as herbertsmithite [33], volborthite [34, 35], or vesignieite [36] for realizing AFHK. More importantly for our present purpose, it has been suggested that kagome compounds with breathing anisotropy, i.e., different Heisenberg exchange coupling on the upward ($J_\triangle$) and downward triangles ($J_\triangledown$) can still host QSL ground-states even at relatively large breathing anisotropy [37]. Signatures of a gapless spin-liquid have recently been observed experimentally in a vanadium oxyfluoride compound, $[NH_4]_2[\mathbf{C_7H_{14}N}][\mathbf{V_7O_6F_{18}}]$ [37–39], at $J_\triangledown/J_\triangle \approx 0.55$. This finding is of particular importance since naturally kagome compound tend to be anisotropic in nature due to impurities or effective perturbations which influence the strength of coupling on different kagome triangles. The spin-$\frac{1}{2}$ breathing-kagome Heisenberg model (BKH) of Eq.(1) has first been studied in the nineties [14,40]. An effective model has been derived in terms of two local degrees of freedom per strong triangle, and a mean-field decoupling of these degrees of freedom has led to the conclusion that a family of short-range resonating valence bond (RVB) states could provide a good variational basis [14,41]. Following up on the recent experimental interest in that system, several theoretical studies [42–44] aiming at providing further insight into the experimental results have revisited that model. While an earlier VMC study predicted a stabilized $\mathbb{Z}_2$ gapped spin-liquid ground-state for the BKH model for $J_\triangledown/J_\triangle \approx 0.7$ [43], other studies based on DMRG on semi-infinite cylinders [44] and more refined VMC calculations [42] predicted the persistence of the gapless U(1) spin-liquid up to large breathing anisotropy. The DMRG results further provide evidence that the BKH model undergoes a phase transition to a nematic phase at $J_\triangledown/J_\triangle \approx 0.2$ or below [44]

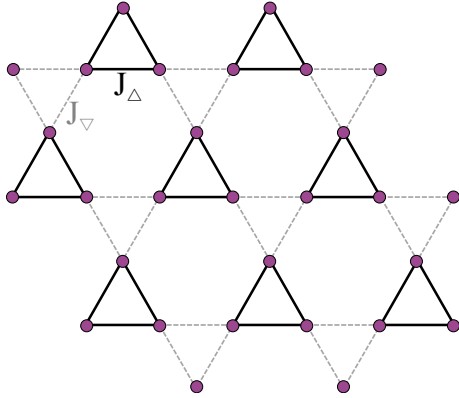

Figure 1: (Color online) Kagome lattice with anisotropic breathing interactions. The AF Heisenberg exchange couplings are different on the edges of upward and downward triangles.

while the VMC results are in favor of a valence-bond-crystal (VBC) which is stabilized for $J_{\bigtriangledown}/J_{\bigtriangleup} \lesssim 0.25$ [42]. Note that, in this parameter range, a simplex $\mathbb{Z}_2$ RVB state represented as a simple TN was found to have a slightly higher energy [42].

Motivated by the importance of recent experiments on vanadium compounds [37–39] and the quest for experimental realization of QSL phases, we address in this paper the spin-$\frac{1}{2}$ breathing-kagome Heisenberg problem and study, in detail, the full phase diagram and ground-state properties of the BKH model in the thermodynamic limit, by resorting to large-scale tensor network calculations based on infinite projected entangled-pair state (iPEPS) [45–48] and infinite projected entangled-simplex state (PESS) [28, 32] methods on the infinite 2D kagome lattice. In particular, we focus more on the large breathing anisotropy limit and perform accurate energy analysis and finite-size entanglement scaling of energies to try and reveal the true ground-state of the system out of the energetically competing VBC, $\mathbb{Z}_2$ QSL and nematic phases. Our results suggest that the U(1) QSL phase of the isotropic kagome Heisenberg antiferromagnet is stable up to very large breathing anisotropy $J_{\bigtriangledown}/J_{\bigtriangleup} \approx 0.05$ and that, for larger anisotropy, it undergoes a first-order quantum phase transition (QPT) to a critical lattice-nematic phase. We capture the lattice-nematic ordering by accurate analysis of the energy density on every bond of the up and down triangles of the kagome lattice in translationally invariant unit-cells with different sizes and further reveal the critical nature of the lattice-nematic phase showing in particular power-law spin-spin correlations along the emerging chains.

The paper is organized as follows. In Sec. 2 we introduce the BKH model on the kagome lattice and briefly discuss the details of the iPEPS and PESS machinery we used for evaluating the ground-state of the system. Next, in Sec. 3 we elaborate on the the U(1) gapless QSL phase of BKH model at the isotropic point with no breathing anisotropy. We further study the large breathing anisotropic limit of the BKH model and lattice-nematic ground-state of the system in Sec. 4. In Sec. 5 we investigate the quantum phase transition and full phase diagram of the BKH model. Finally Sec. 6 is devoted to a discussion and to a conclusion.

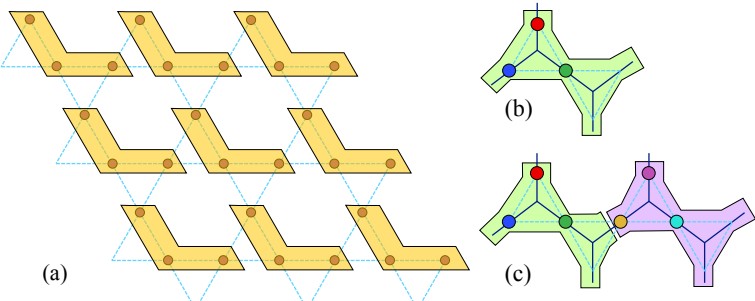

Figure 2: (Color online) (a) Coarse-graining the kagome lattice to a square network of block sites with three spin-$\frac{1}{2}$ in each block and a total local Hilbert space of $d = 2^3 = 8$. The iPEPS simulations were performed on $2 \times 2$ and $4 \times 4$ unit-cells of block-sites with 12 and 48 spins, respectively. (b) Coarse-graining of a PESS with three physical tensors and two simplices into a block-site. (c) Coarse-graining of a PESS with six physical tensors and four simplices into block-sites.

## 2  Model and Method

The spin-$\frac{1}{2}$ breathing-kagome Heisenberg antiferromagnetic model [14, 40] is defined by

$$H = J_\triangle \sum_{\langle ij \rangle \in \triangle} \mathbf{S}_i \cdot \mathbf{S}_j + J_\triangledown \sum_{\langle ij \rangle \in \triangledown} \mathbf{S}_i \cdot \mathbf{S}_j, \tag{1}$$

where the first (second) sum runs over edges of the upward, $\triangle$, (downward, $\triangledown$) triangles of the kagome lattice (see Fig. 1). Here $J_\triangle$ and $J_\triangledown$ are the antiferromagnetic (AF) Heisenberg exchange couplings, respectively on the up and down triangles and $\mathbf{S}_i$ is the spin operator at lattice site $i$. As discussed above, we are interested in analyzing the ground-state properties of Hamiltonian (1) for the full range of the $J_\triangledown/J_\triangle$ coupling. Here without loss of generalities, we set $J_\triangle = 1$.

In order to study the ground-state of the BKH model, we used two different tensor network methods based on infinite projected entangled-pair state [45–48] and infinite projected entangled-simplex state [28, 32]. The iPEPS algorithm was applied to the BKH problem by coarse-graining the kagome lattice to a square TN of block-sites (each block with three spin-$\frac{1}{2}$) of rank-5 tensors with physical bond dimension $d = 2^3 = 8$ and virtual bond dimension $D$ [13, 49–51]. The coarse-grained lattice is illustrated in Fig. 2-(a). In order to approximate the ground-state of the BKH model, we used both simple-update (SU) [26, 50, 52] and full-update (FU) [13, 53–55] based on imaginary-time evolution (ITE) [47, 56] accompanied by corner transfer matrix renormalization group (CTMRG) [48, 52, 55] for the contraction of the infinite 2D lattice, applied to $2 \times 2$ and $4 \times 4$ unit-cells of block-sites with 12 and 48 spins, respectively. The PESS, however, was performed using 3PESS machinery [32] with rank-3 simplex tensors with no physical leg, which captures the three-partite entanglement inside kagome triangles, and rank-3 physical tensors representing local spins. The contraction of the 2D PESS tensor network was performed by first coarse-graining two simplices and three physical tensors into a block-site with local Hilbert space of $d = 2^3 = 8$ and then using the CTMRG. We used two different unit-cells for the PESS simulations, one with three physical tensors and two simplices (Fig. 2-(b)) and another one with six physical tensors and four

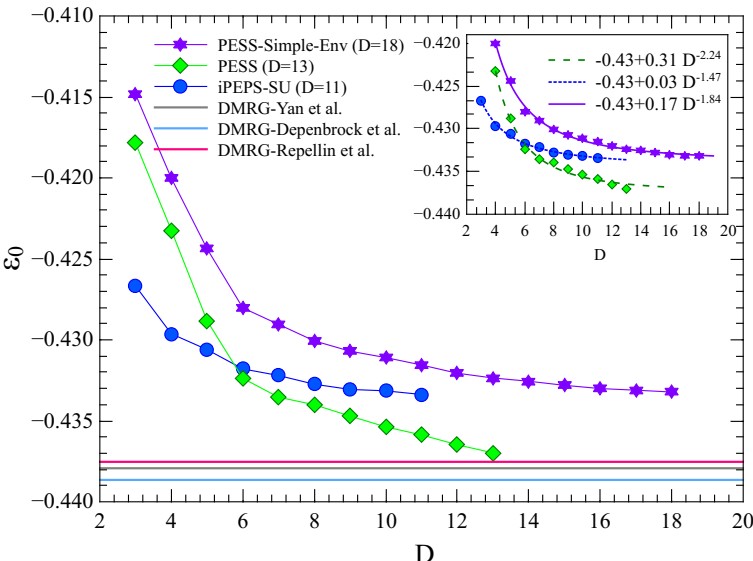

Figure 3: (Color online) Scaling of the ground-state energy, $\varepsilon_0$, with respect to bond dimension $D$ at the isotropic point, $J_{\bigtriangledown}/J_{\bigtriangleup} = 1$, for iPEPS and PESS for both full and simple calculation of environment. The inset shows the power-law extrapolation of the iPEPS and PESS energies with $\varepsilon_0(D) = \varepsilon_0 + aD^{-\beta}$ form. The $a$ and $\beta$ parameters for each curve are given in the plot legend. The solid lines further show the results from Refs. [11, 25, 44].

simplices (Fig. 2-(c)). While both unit-cells are suitable for capturing U(1) QSL, $\mathbb{Z}_2$ RVB state and lattice-nematic phases, the VBC ordering is only commensurate with the latter unit-cell [42]. We further used simple update based on ITE and high-order singular value decomposition (HOSVD) [32] to optimize the PESS tensors.

The ITE iterations for both iPEPS and PESS methods were performed starting from $\delta\tau = 10^{-1}$ down to $10^{-5}$ with at least 4000 iterations for each $\delta\tau$. We further initialized the simulations with different initial tensors ranging from totally random states to a $D = 3$ RVB state for different regimes of the problem. Moreover, for each bond dimension $D$, we recycled both ground-state tensors and the environment from the previous lower bond. In order to check the convergence of the algorithms, we compared the energy difference of two consecutive iterations with a tolerance threshold of $10^{-14}$ for each ITE $\delta\tau$. In the end, the expectation values of local operators and variational energies were calculated by using the converged ground-state tensors and full environment tensors obtained from CTMRG algorithms [52, 54].

Let us further note that within our available resources and time, we managed to push the iPEPS simulations up to $D = 6$ for the full-update and $D = 11$ for the simple-update. The PESS simulations were also performed up to $D = 13$ with the full calculation of the environment, i.e., with CTMRG and up to $D = 18$ for the simple environment, i.e., with $\lambda$ matrices of the HOSVD optimization. However, since for the simple environment the energies are no longer variational we do not show them here. Nevertheless, the results were consistent with full environment calculations. Moreover, we fixed the boundary dimension to $\chi = 80$ for $D \leq 8$ and $\chi = 64$ for $D > 8$ in both the iPEPS and PESS simulations. Whenever possible, we further checked the convergence of the simulations for $\chi > 64$ for various $D$ values, and we have noticed no significant change in energies for $\chi > 64$ in our results.

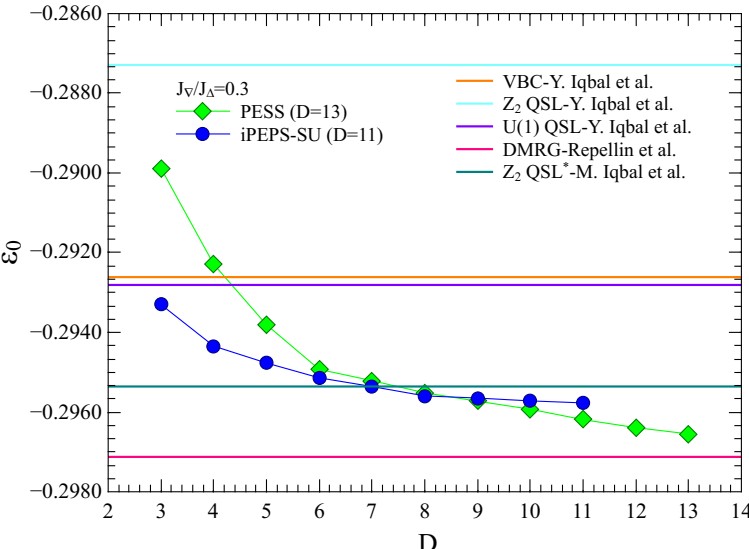

Figure 4: (Color online) Scaling of the ground-state energy, $\varepsilon_0$, with respect to bond dimension $D$ at $J_{\bigtriangledown}/J_{\bigtriangleup} = 0.3$. The power law scaling of energies with respect to $D$ is similar to the isotropic point, $J_{\bigtriangledown}/J_{\bigtriangleup} = 1$, suggesting that both points belong to the same phase.

In order to avoid confusion between different results obtained from different methods, we fix the following style in the figures throughout the paper: The iPEPS data obtained with simple- (full-) update are demonstrated with circles (squares) and the PESS data are depicted with diamonds. We have further used stars for illustrating non-variational PESS energies which are obtained from $\lambda$ matrices of the HOSVD optimization.

In the following, we analyze the BKH Hamiltonian (1) for all ranges of AF Heisenberg exchange coupling and map out the full phase diagram of the system. More precisely, we calculate the ground-state of the BKH model for $0 \leq J_{\bigtriangledown}/J_{\bigtriangleup} \leq 1$ and detect possible phase transitions by following the ground-state energy per site, $\varepsilon_0$, and its derivative, as well as two-point spin correlators. However, before we start analyzing the phase diagram of the BKH model, let us first consider the extreme limits of Hamiltonian (1).

# 3 Small Breathing Anisotropy

In the limit of the BKH model where $J_{\bigtriangledown}/J_{\bigtriangleup} = 1$, i.e., the isotropic point, it has already been shown that tensor network simulations favor a gapless U(1) QSL with algebraic decay of correlations. We refer the interested reader to Refs. [27–29, 32] for detailed discussions on the isotropic spin-$\frac{1}{2}$ kagome Heisenberg antiferromagnet. Nevertheless, in order to check the accuracy and efficiency of our algorithms, we applied both iPEPS and PESS methods to the BKH model at the isotropic point and recovered the results of Ref. [28, 32].

Fig. 3 illustrates finite-entanglement scaling of the iPEPS and PESS energies for both simple and full calculation of the environment with respect to bond dimension $D$. We find that $\varepsilon_0$ converges algebraically with $D$ with power-law form $\varepsilon_0(D) = \varepsilon_0 + aD^{-\beta}$, as shown in the inset of the figure for both iPEPS and PESS energies. The power-law convergence of energy

with respect to bond dimension has already been suggested as a good numerical signature of critical (gapless) systems, both in one [56] and two [27, 28] dimension. Our energies obtained with both iPEPS and particularly with PESS shows the algebraic convergence, consistent with a gapless ground state. Besides, we have contrasted the power-law decay of energies, $\varepsilon_0(D) = \varepsilon_0 + aD^{-\beta}$, against exponential, $\varepsilon_0(D) = \varepsilon_0 + \exp(-D)$ and logarithmic decay, $\varepsilon_0(D) = \varepsilon_0 + \log(D)$ (not shown in the paper) and found that the converges of our energies for both PESS and iPEPS simulations are best fitted with a power-law decay signaling a gapless underlying state.

We have further analyzed the energies on the upward and downward triangles of the kagome lattice as well as correlations $\langle S_i^\alpha . S_j^\alpha \rangle$ ($\alpha = x, y, z$) on every link of each triangle in the unit-cell and and found the same energies and correlations on both triangles with a uniform ground-state which respects all lattice symmetries and SU(2) symmetry of the Hamiltonian (1), thus indicating the QSL nature of the ground-state at the isotropic point.

Our best variational energy at the isotropic point is $\varepsilon_0 = -0.436979$, obtained from PESS with $D = 13$ which is lower than that of Ref. [32] for $D = 13$ 9-PESS. Let us further note that our iPEPS energy at this point is $\varepsilon_0 = -0.433374$ for $D = 11$, which is slightly higher compared to that of the PESS due to the lower maximum achievable bond dimension. Besides, it is already known that the simplex structure of the PESS tensors can capture the three-partite entanglement inside a frustrated kagome triangle better and, hence, is able to yield lower energies compared to iPEPS which only captures bipartite entanglement on bonds of the lattice [28, 32].

Let us further note that characterizing the U(1) or $\mathbb{Z}_2$ nature of the state is a numerically nontrivial task. The gapless spin liquid is expected to have long-ranged entanglement and correlation functions and the U(1) state has no well-characterized topological order. We calculated the reduced density matrix of the ground state obtained from our PESS simulation on bipartitions of a semi-infinite cylinder and calculated the entanglement entropy [57]. We found no topological contribution to the entanglement entropy which is in agreement with that of a U(1) gapless spin liquid. Moreover, the momentum-dependent excitation spectrum which was extracted from the DMRG transfer matrix of Ref. [29, 44], exhibits Dirac cones that match those of a $\pi$-flux free-fermion model (the parton mean-field ansatz of a U(1) Dirac spin liquid). Although we are not able to calculate the same information within the framework of our TN simulations, we provided evidences through the paper that our results are in agreement with these DMRG calculations of Ref. [44] and therefore we believe that the gapless state we capture at the isotropic point should be a U(1) spin liquid.

Immediately away from the isotropic point, i.e., for $J_\triangledown / J_\triangle < 1$, the difference in the $J_\triangle$ and $J_\triangledown$ couplings in Hamiltonian (1) breaks the lattice inversion center (i.e. the $180^o$ rotational symmetry). However, as long as the breathing anisotropy is not too large, we expect the QSL to remain stable. Our analysis indeed confirms that introducing small breathing anisotropy does not destroy the QSL ground state of the spin-$\frac{1}{2}$ kagome Heisenberg antiferromagnet and the uniformity and SU(2) invariance of the ground-state are preserved at the level of each individual triangles. Nevertheless, upward triangles will have lower energies due to larger $J_\triangle$ couplings compared to $J_\triangledown$. Fig. 4 shows the ground-state energy of the system for both iPEPS and PESS at $J_\triangledown / J_\triangle = 0.3$. The power law scaling of energies with respect to $D$ is similar to the isotropic point, $J_\triangledown / J_\triangle = 1$, which suggests that both points belong to the same phase. In fact, in future sections, and especially in Fig. 8, we will show that the U(1) QSL ground-state of the BKH model at the isotropic point persists to very large breathing anisotropies.

We postpone further discussions regarding the stability and persistence of the U(1) QSL

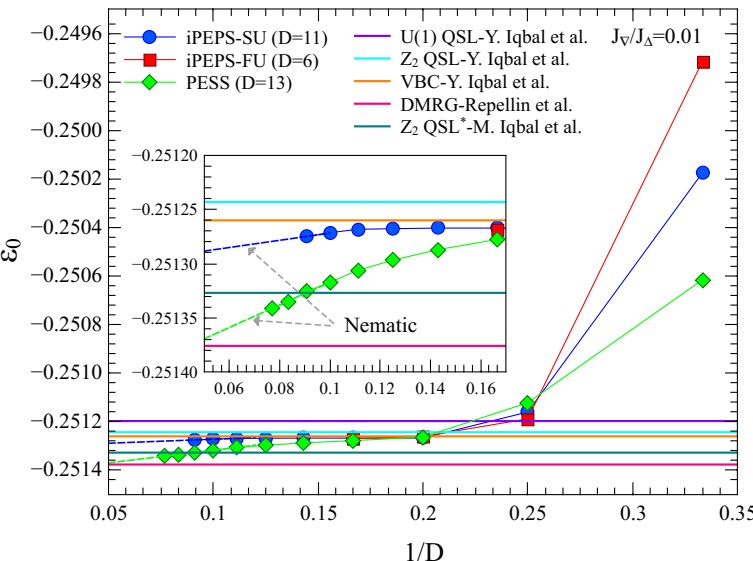

Figure 5: (Color online) Scaling of the iPEPS and PESS ground-state energy per-site with respect to inverse bond dimension $D$ in the large breathing anisotropy limit at $J_\bigtriangledown/J_\bigtriangleup = 0.01$. Our variational energies obtained with SU, FU and PESS are lower as compared to those of Ref. [42] for U(1) QSL, $\mathbb{Z}_2$ QSL and VBC phases and are in agreement with the DMRG results of Ref. [44,58] for a lattice-nematic phase.

ground-state of the system in the presence of breathing anisotropy to Sec. 5.

## 4   Large Breathing Anisotropy

In this section, we elaborate on the less studied limit $J_\bigtriangledown/J_\bigtriangleup \ll 1$, i.e. the large breathing anisotropic regime. In the extreme case where the couplings on the down triangles are zero, $J_\bigtriangledown = 0$, the system is composed of decoupled upward triangles with AF interactions, with a highly-degenerate ground-state and a ground-state energy $\varepsilon_0 = -0.25$.

By switching on and gradually increasing the $J_\bigtriangledown$ couplings on the downward triangles, forming an ordered ground-state becomes a highly non-trivial task. An early study based on the short-range RVB basis [41] suggests that a gapped $\mathbb{Z}_2$ spin-liquid ground-state may emerge in the presence of large breathing anisotropy. In another study based on Gutzwiller projected wave functions, the analysis of the energy shows that the U(1) spin-liquid phase of the BKH model undergoes a dimer instability at large breathing anisotropy and a VBC phase is stabilized for $J_\bigtriangledown/J_\bigtriangleup < 0.25$ [42]. By contrast, recent DMRG study on semi-infinite cylinders suggest a lattice-nematic ordering for the ground-state of the BKH model at large breathing anisotropies [44] which emerges around $J_\bigtriangledown/J_\bigtriangleup \approx 0.14$.

In order to further investigate the nature of the ordering of the system at large breathing anisotropies, we performed systematic energy analysis for the whole range of couplings. In particular, we performed accurate entanglement-scaling of energies versus inverse bond dimension, $1/D$ up to $D = 13$ for several points at very large anisotropies. Fig. 5 depicts the scaling of the ground-state energy versus inverse bond dimension, $1/D$, for $J_\bigtriangledown/J_\bigtriangleup = 0.01$.

Table 1: Variational energies obtained from iPEPS and PESS compared with U(1), $\mathbb{Z}_2$ QSL and VBC energies of Ref. [42] and DMRG data of Ref. [44,58] for YC8-2 cylinder geometry. $\mathbb{Z}_2$ QSL* is an improved RVB ansatz [59]. The lowest possible energies for all couplings belong first to the DMRG results and next to our PESS ($D = 13$) simulations.

| $J_\bigtriangledown/J_\bigtriangleup$ | iPEPS-FU ($D = 6$) | iPEPS-SU ($D = 11$) | PESS ($D = 13$) | U(1) QSL | $\mathbb{Z}_2$ QSL | $\mathbb{Z}_2$ QSL* | VBC | DMRG |
|---|---|---|---|---|---|---|---|---|
| 0.01 | −0.251268 | −0.251275 | −0.251363 | −0.251197 | −0.251243 | −0.251327 | −0.251260 | −0.251376 |
| 0.03 | | | −0.254092 | −0.253641 | −0.253729 | −0.254017 | −0.253815 | −0.254195 |
| 0.05 | −0.256801 | −0.256816 | −0.256893 | −0.256147 | −0.256215 | −0.256755 | −0.256215 | −0.257079 |
| 0.07 | | | −0.259818 | −0.258717 | −0.258701 | −0.259545 | −0.259057 | −0.260015 |
| 0.09 | | | −0.262697 | −0.261349 | −0.261187 | −0.262386 | −0.261744 | −0.262994 |
| 0.1 | | −0.264146 | −0.264138 | −0.262690 | −0.262430 | −0.263827 | −0.263104 | −0.264498 |
| 0.2 | | −0.279362 | −0.279568 | −0.276960 | −0.274860 | −0.278971 | −0.277309 | −0.280211 |
| 0.3 | | −0.295770 | −0.296546 | −0.292810 | −0.287290 | −0.295364 | −0.292613 | −0.297125 |
| 0.4 | | −0.313191 | −0.314549 | −0.310240 | −0.299720 | −0.312873 | −0.309018 | −0.316911 |
| 0.5 | | −0.331635 | −0.333410 | −0.329250 | −0.312150 | −0.331304 | −0.326522 | −0.333897 |
| 1.0 | | −0.433374 | −0.436979 | | −0.374300 | −0.433325 | −0.430545 | −0.437533 |

Our energies obtained with both full-update (FU) and simple-update (SU) of the iPEPS as well as PESS results with full calculation of the environment are reported in Table 1 and compared to the results of Ref. [42,59] for U(1), $\mathbb{Z}_2$ QSL and VBC as well as the DMRG results of Ref. [44,58]. Our best variational energy is $\varepsilon_0 = -0.251363$ which was obtained from PESS ($D = 13$). Our energies are also in agreement with DMRG results of Ref. [44,58] which is slightly lower than our iPEPS and PESS results.

Next, in order to identify the nature of the instability in the large breathing anisotropy limit, we analyzed the energy density and nearest-neighbor spin-spin correlations on every bond of the upward and downward triangles of the kagome unit-cells (see Fig. 2) with different structures. We observed that energies and correlations on different links of the up and down triangles of the kagome lattice undergo a dimer instability with broken $C_3$ rotational symmetry at the level of individual triangles which are consistent with the lattice-nematic pattern.

Fig. 6 illustrates the lattice-nematic pattern with strong and weak correlation on the bonds of the kagome lattice that we captured within the framework of our TN simulations in the large breathing anisotropy limit at $J_\bigtriangledown/J_\bigtriangleup = 0.01$ for all bond dimensions $D$ and unit-cell structures. Let us further point out that depending on the choice of the initial states, size of the unit-cells and the virtual bond dimension, we observed a spontaneous breaking of mirror symmetry on the up triangles in our TN simulations. The broken mirror symmetry was detected from the difference in the correlations on the two green links of up triangles, $\bigtriangleup_1, \bigtriangleup_2$, shown in Fig. 6. We observed that this difference in the correlations of the $\bigtriangleup_1, \bigtriangleup_2$ links decreases upon increasing the bond dimension, and that the mirror symmetry is restored in the thermodynamic limit. The same behaviour has already been observed in the DMRG simulations of Ref. [44,58].

The nematic state breaks the $C_3$ local rotational symmetry of the triangles while preserving the translational invariance in every direction of the lattice. This is in contrast with the VBC phase which breaks both translational and rotational symmetries of the system [42]. The lattice-nematic state on the kagome lattice is, in fact, three-fold degenerate. Repeating the simulations with different initial states, we found the two other degenerate nematic states with

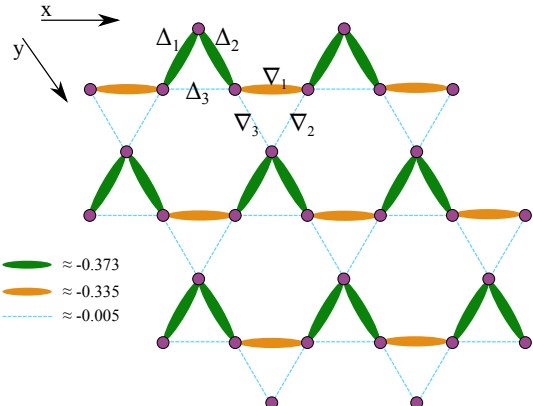

Figure 6: (Color online) The lattice-nematic pattern in the large breathing anisotropy limit at $J_{\triangledown}/J_{\triangle} = 0.01$ constructed from strong and weak correlations on the links of the upward and downward kagome triangles which breaks the $C_3$ local rotational symmetry of the system while preserving the translational invariance in every direction on the lattice. The correlations reported in the figure are obtained with PESS $D = 13$. The values on the green links are the average correlation of the two edges (see the text for further discussion).

the same magnitude of correlations on bonds but a different pattern. This implies that in the regime of very large breathing anisotropy, the system undergoes a dimensional reduction with three degenerate ground states that consist of almost decoupled chains. Let us further stress that our results for the large breathing anisotropies are in agreement with the recent DMRG results [44].

For completeness of our analysis and further to see how the lattice-nematic state competes with other states from previous studies, we consider the Taylor expansion of the BKH energy in the large breathing anisotropy limit up to second order in perturbation theory

$$\frac{\varepsilon_0}{J_{\triangle}} = -0.25 + c_1 \frac{J_{\triangledown}}{J_{\triangle}} + c_2 \left(\frac{J_{\triangledown}}{J_{\triangle}}\right)^2 + \dots . \tag{2}$$

The first constant term in the above equation is the energy of decoupled upward triangles. The coefficients $c_1$, $c_2$ can further be obtained by quadratic fits of the energy curve.

Table 2 provides the $c_1$, $c_2$ expansion coefficients of our lattice-nematic states obtained with PESS ($D = 13$) compared with those of the effective model known as trimerized kagome model, which corresponds to a frustrated spin-orbital model on the triangular lattice [14,44]. Note that, since our energy curve shows a discontinuity in its slope, we have performed two separate fits on each side of the discontinuity. The coefficients for $\mathbb{Z}_2$, U(1) QSLs and VBC state of Ref. [42] and $\mathbb{Z}_2$ QSL* of Ref. [59] as well as those of the nematic state obtained with DMRG in Ref. [44] are also provided in the table. One can clearly see that the $c_1$ coefficient of the nematic state obtained both in our simulations and previous DMRG results is larger (in absolute value) than those of the $\mathbb{Z}_2$, U(1) QSLs and VBC states suggesting a stabilized nematic state as the true ground state of the BKH model in the large breathing anisotropy limit.

We have further calculated the long-range spin-spin correlation defined by $C(r) = \langle \mathbf{S}_{(x,y)}.\mathbf{S}_{(x+r,y)}\rangle - \langle \mathbf{S}_{(x,y)}\rangle.\langle \mathbf{S}_{(x+r,y)}\rangle$ in the large breathing anisotropy limit compared with several points in the

Table 2: The $c_1$, $c_2$ coefficients of the Taylor expansion for the BKH model at large breathing limit obtained with PESS ($D = 13$) compared with those of the DMRG data for the effective model in Ref. [44, 58] (after extrapolation to infinite cylinders) as well as with the $\mathbb{Z}_2$, U(1) QSLs and VBC states of Ref. [42] and $\mathbb{Z}_2$ QSL* of Ref. [59].

| Wave Function | $c_1$ | $c_2$ |
|---|---|---|
| nematic (PESS) | $-0.1354$ | $-0.0537$ |
| U(1) QSL (PESS) | $-0.1347$ | $-0.0670$ |
| U(1) QSL [42] | $-0.1190$ | $-0.079$ |
| $\mathbb{Z}_2$ QSL [42] | $-0.1245$ | $0$ |
| VBC [42] | $-0.1255$ | $-0.055$ |
| $\mathbb{Z}_2$ QSL* [59] | $-0.1323$ | $-0.0628$ |
| Effective Model [44] | $-0.1353$ | $0$ |

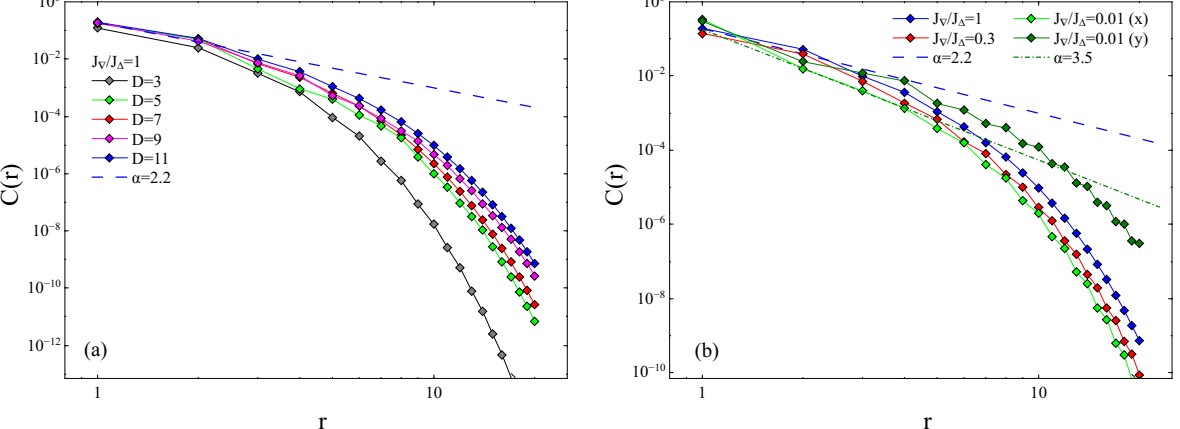

Figure 7: (Color online) Log-log plot of the long range spin-spin correlations $C(r) = \langle \mathbf{S}_{(x,y)} . \mathbf{S}_{(x+r,y)} \rangle$ obtained with PESS (a) at the isotropic point, $J_{\triangledown}/J_{\triangle} = 1$, for different $D$ and (b), the $C(r)$ for PESS ($D = 11$) in the nematic phase at large breathing anisotropy $J_{\triangledown}/J_{\triangle} = 0.01$ compared with those of the QSL phase at $J_{\triangledown}/J_{\triangle} = 0.3$ and 1. All correlations show approximate power-law decays, $C(r) \sim C_0 r^{-\alpha}$, shown by straight dashed lines. A similar decay up to 7 or 8 lattice spacings is observed in the QSL phase and along the chain of the nematic phase, with an exponent $\alpha \simeq 2.2$. The green dashed line shows the power-law fit in the nematic phase, perpendicular to the chains, with $\alpha \simeq 3.5$.

small-breathing regime. Fig. 7-(a) shows the log-log plot of $C(r)$ obtained with PESS for different bond dimensions at the isotropic point, $J_\bigtriangledown/J_\bigtriangleup = 1$. The $C(r)$ at small distances shows a power-law behaviour with a tail which deviates from the power-law fit (dashed line) at large distances. However, by increasing the bond dimension, $D$, the spin-spin correlation tends to approach the power-law fit indicating a gapless ground-state for the isotropic point, at the thermodynamic limit. Fig. 7-(b) further demonstrates the $C(r)$ obtained with PESS ($D = 11$) at $J_\bigtriangledown/J_\bigtriangleup = 0.01$ in the lattice-nematic phase and $J_\bigtriangledown/J_\bigtriangleup = 0.3$ compared with the isotropic point in the spin-liquid phase. $C(r)$ in the QSL phase behaves similarly in different directions of the lattice, as expected from a QSL phase with no broken symmetry. Most importantly, $C(r)$ for $J_\bigtriangledown/J_\bigtriangleup = 0.3$ and 1 decay similarly especially at small distances, suggesting that they belong to the same phase. This is another strong signature that the QSL phase persists in the large breathing anisotropy limit. However, $C(r)$ in the lattice-nematic phase is different along the strong chain, $x$-direction, and perpendicular to the chain, $y$-direction, as shown in Fig. 7-(b). We have therefore approximated the power-law decay of $C(r)$ at $J_\bigtriangledown/J_\bigtriangleup = 0.3$ and 1 and $J_\bigtriangledown/J_\bigtriangleup = 0.01$ along the strong chain with $C(r) \sim C_0 r^{-\alpha}$, shown as a dashed blue line in the Fig. 7-(b). $C(r)$ decays algebraically with $C_0 = 0.18, \alpha = 2.2$ indicating that the wave functions at $J_\bigtriangledown/J_\bigtriangleup = 0.3$ and 1 and at $J_\bigtriangledown/J_\bigtriangleup = 0.01$ along the strong chain are critical. The spin-spin correlation perpendicular to the strong chain at $J_\bigtriangledown/J_\bigtriangleup = 0.01$ shows a different power-law decay with $C_0 = 0.18, \alpha = 3.5$, which is the expected behavior of the lattice-nematic state with a different pattern in the $x$- and $y$-direction.

Note that our findings for the large breathing anisotropy limit are in agreement with the Lieb-Schultz-Mattis theorem [60, 61] which states that, for systems with half-odd integer spins in the unit-cell, there cannot exist a gapped spin-liquid with a unique ground state. Therefore, the critical nature found in both phases in the phase diagram is consistent with the theorem.

Let us note that our TN ansatz suffers from a small spurious magnetic ordering of the order $\sim 0.003$ which is believed to be an artifact of the PEPS methods with finite bond dimension. The effects of such a spurious magnetization is then seen as zigzag oscillations in $\langle \mathbf{S}_{(x,y)} . \mathbf{S}_{(x+r,y)} \rangle$. We, therefore, subtract the local magnetic contribution from the spin-spin correlation to correctly capture the power-law decay.

## 5    Quantum Phase Transition

In previous sections, we have identified and characterized two different phases at the extreme regimes of the BKH Hamiltonian (1), the U(1) spin-liquid at the isotropic point $J_\bigtriangledown/J_\bigtriangleup = 1$ with zero anisotropy and the lattice-nematic phase at large breathing anisotropy limit, $J_\bigtriangledown/J_\bigtriangleup \ll 1$. It is, therefore, reasonable to expect a quantum phase transition between the two extreme phases. In order to study the QPT, we analyzed the whole regime of the parameter space, $0 \leq J_\bigtriangledown/J_\bigtriangleup \leq 1$, and calculated the ground-state energy of the BKH model for different bond dimensions. Fig. 8-(a) shows the ground-state energy of the PESS simulations for the whole range of couplings (up to $D = 13$) and its first-order derivative (see the inset). The sharp discontinuity in the derivative of the energy reveals a first-order quantum phase transition at $J_\bigtriangledown/J_\bigtriangleup \approx 0.05$ between the U(1) QSL and the lattice-nematic phase of the BKH model. Fig. 8-(b) further depicts the evolution of the location of the transition point versus $1/D$ indicating that within our finite $D$ PESS simulations the location of the transition point does not seem to change with increasing the bond dimension for $D \geq 11$.

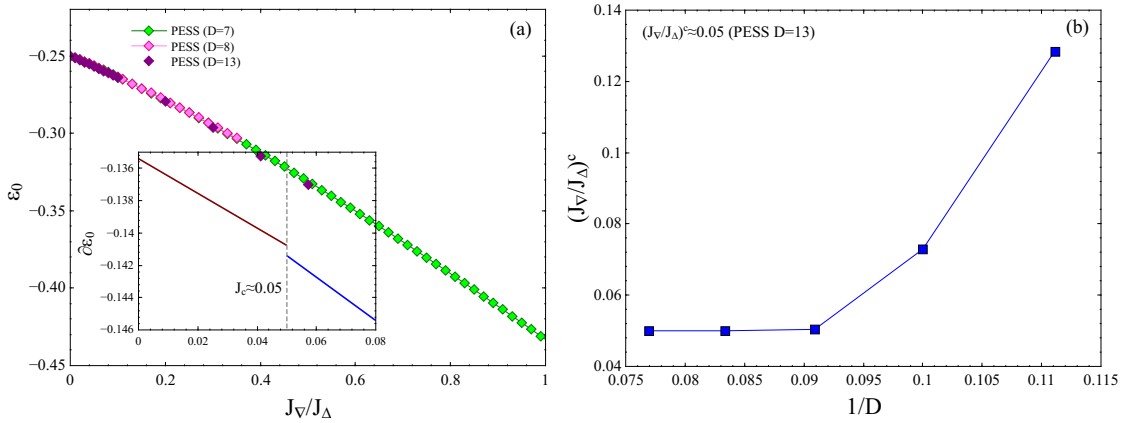

Figure 8: (Color online) (a) Ground-state energy per-site of the BKH model for $0 \le J_\bigtriangledown/J_\bigtriangleup \le$ 1. The U(1) QSL persists to large breathing anisotropies and breaks down to a lattice-nematic phase at $J_\bigtriangledown/J_\bigtriangleup \approx 0.05$. The inset shows the derivative of the energy series, $\partial\varepsilon_0 = c_1 + 2c_2(J_\bigtriangledown/J_\bigtriangleup)$, obtained with $c_1$ and $c_2$ coefficients of the nematic and U(1) QSL phases from Table 2 for $D = 13$. The sharp discontinuity in the derivative is a clear signature of the first-order phase transition. (b) Scaling of the location of the transition point with inverse bond dimension, $1/D$.

In an attempt to draw conclusions about the infinite $D$ limit, we have kept track of the evolution of the coefficients $c_1$ and $c_2$ with $1/D$. First of all, we have estimated the error bars on these coefficients. Fig. 9-(a-d) shows the fitting of PESS energies ($D = 13$) using the $c_1$ and $c_2$ coefficients of Table 2. The error bars on the energies, estimated from the standard deviation from the mean energy values, are very small, of the order $10^{-5} - 10^{-6}$ for different couplings, but they still lead to a non negligible error threshold of the order $10^{-3}$ on the $c_1$ and $c_2$ coefficients. The resulting estimates for $c_1$ are plotted as a function of $1/D$ in Fig. 10, together with the DMRG results of Ref. [44,58] for the effective first-order model in the left panel, and with those of the RVB wave function of Ref. [59] in the right panel. Several remarks are in order. First of all, the values of $c_1$ for the nematic PESS and for DMRG are very close to each other, a nice confirmation that these approaches are indeed describing the same phase. However, this plot also shows the difficulty in extrapolating our results as a function of $1/D$. The naive linear extrapolation suggested by the data if one forgets about the error bars can be excluded because it would lead to a value of $c_1$ much too small as compared to DMRG, and taking the error bars into account leads to a very broad distribution. Besides, there is no theoretical prediction for the evolution of the results as a function of $1/D$ on which to rely to go beyond the naive linear extrapolation. The comparison with DMRG suggests that the coefficients $c_1$ must level off at larger $D$, but we are unfortunately unable to check this. Finally, we note that the coefficient $c_1$ of the $p$-RVB wave-function is not far but above the PESS values for both the nematic and U(1) phases, in agreement with our finding that, at least for finite bond dimension $D$, this phase is not stabilized.

In order to further check the stability of the U(1) QSL on the right side of the QPT point, we performed a careful entanglement scaling of energies for several couplings explicitly for regions previously suggested to host a phase transition, and we investigated the energy densities and bond correlations on all bonds of the upward and downward kagome triangle.

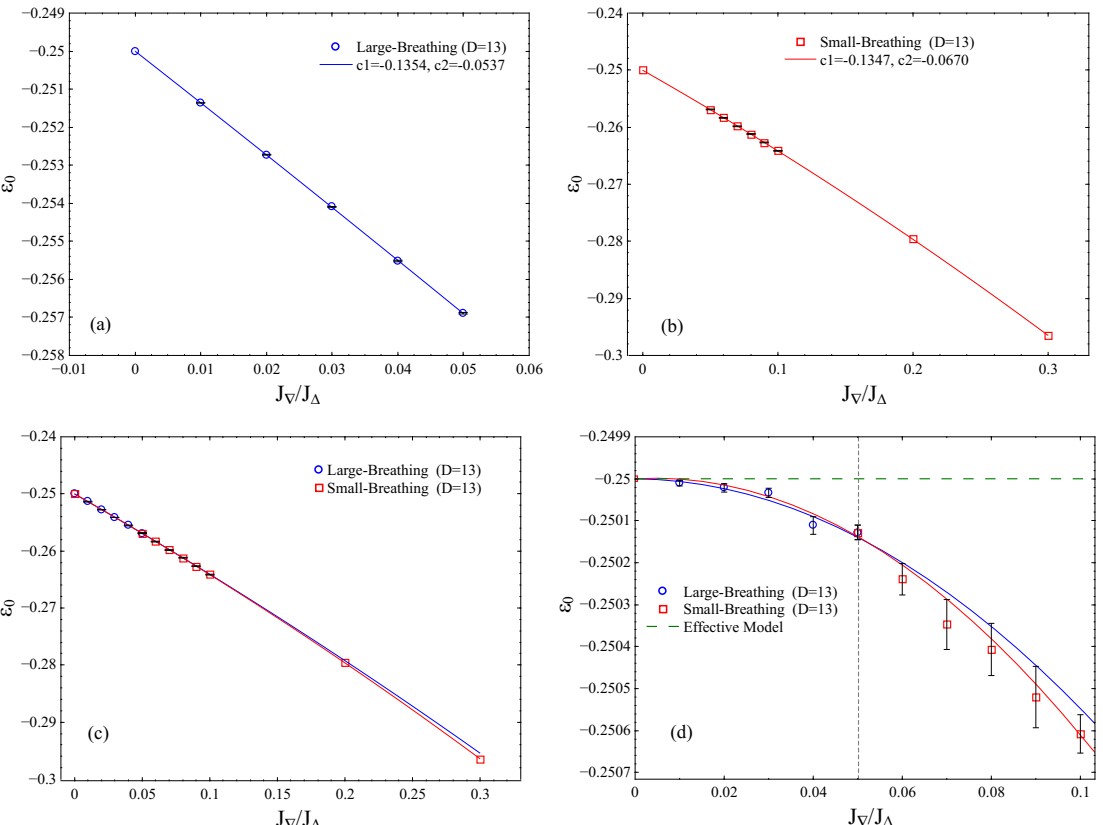

Figure 9: (Color online) Fitting the PESS energies ($D = 13$) in the (a) nematic and (b) U(1) QSL phases using the $c_1$ and $c_2$ coefficients of Table 2. (c) Level crossing of the the small- and large-breathing series. (d) In order to observe the level crossing and the first-order nature of the QPT more appropriately, we have subtracted the linear correction of the effective model from all data points and fits, i.e., $\varepsilon_0 - 0.1353(J_{\triangledown}/J_{\triangle})$. The error bars are standard deviation from the mean-value of PESS data for different simulations.

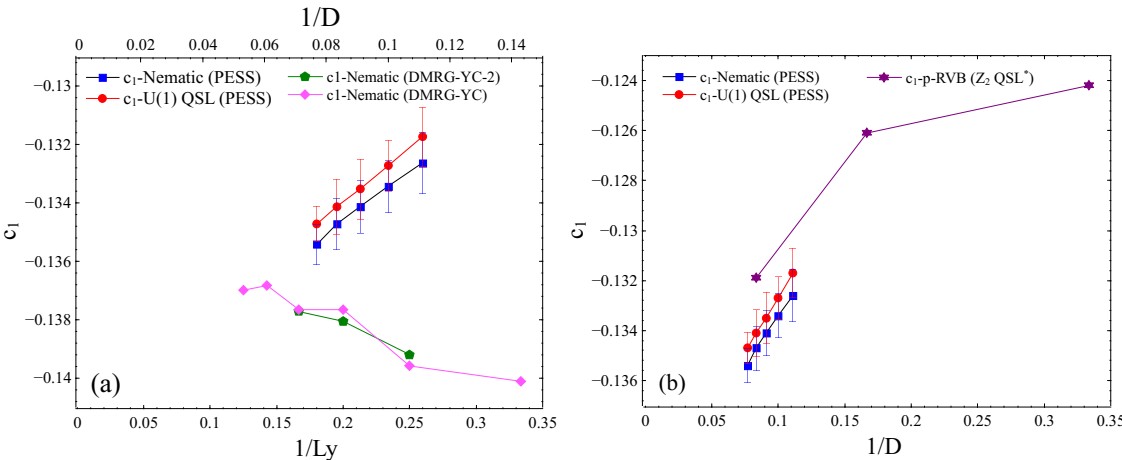

Figure 10: (Color online) Scaling of the PESS $c_1$ coefficient with inverse bond dimension, $1/D$, (a) compared to the scaling of DMRG results of Ref. [44,58] with $1/L_y$ (inverse cylinder width) for different cylinder geometries and (b) compared with the $1/D$ scaling of the p-RVB of Ref. [59] (the wave function for constructing the $\mathbb{Z}_2$ QSL* state). Note that the values for the YC geometry should be considered as upper bounds for the largest values of $L_y$.

Fig. 11 shows the correlations on all links of the up and down kagome triangles for $0 \leqslant J_\bigtriangledown/J_\bigtriangleup \leqslant 1$. One can clearly observe that, in the range $0.1 \leqslant J_\bigtriangledown/J_\bigtriangleup \leqslant 0.5$, the correlations on the links of upwards and on the links of downward triangles are the same, indicating that the $C_3$ rotational symmetry is preserved at the level of each individual triangle. Besides, we performed entanglement scaling of energies for all of the points in Fig. 11 and observed algebraic decay of correlation for all points. Our findings therefore suggest that the algebraic U(1) spin-liquid phase of the spin-$\frac{1}{2}$ kagome Heisenberg antiferromagnet is stable up to very large breathing anisotropies. Our results are in agreement with the recent experiments on vanadium oxyfluoride compounds [37–39] which detected signatures of a gapless U(1) QSL at breathing ratio $J_\bigtriangledown/J_\bigtriangleup \approx 0.55$ [37].

## 6 Discussion and outlook

First introduced as a toy model to study the spin-1/2 kagome antiferromagnet [14], the breathing spin-1/2 kagome antiferromagnet has recently attracted attention on its own due to its experimental relevance for some vanadium compounds [37–39]. In the present paper, we have performed large scale tensor network calculations of that model based on projected entangled-pair state and projected entangled-simplex state methods. The picture emerging from these calculations is consistent with the DMRG results of Ref. [44]: The system seems to be a U(1) liquid from the isotropic limit $J_\bigtriangledown/J_\bigtriangleup = 1$ down to very small values of $J_\bigtriangledown/J_\bigtriangleup$, and to undergo a first-order transition into a critical, lattice nematic phase that breaks rotational symmetry in real space. Our simulations for the largest available values of the bond dimension $D$ locate quite convincingly the transition at $J_\bigtriangledown/J_\bigtriangleup \simeq 0.05$. This should be contrasted to the conclusions of Ref. [44], where the actual critical value of $J_\bigtriangledown/J_\bigtriangleup$ (if any) could not be pinned down because of the still strong dependence of the results on the circumference of the

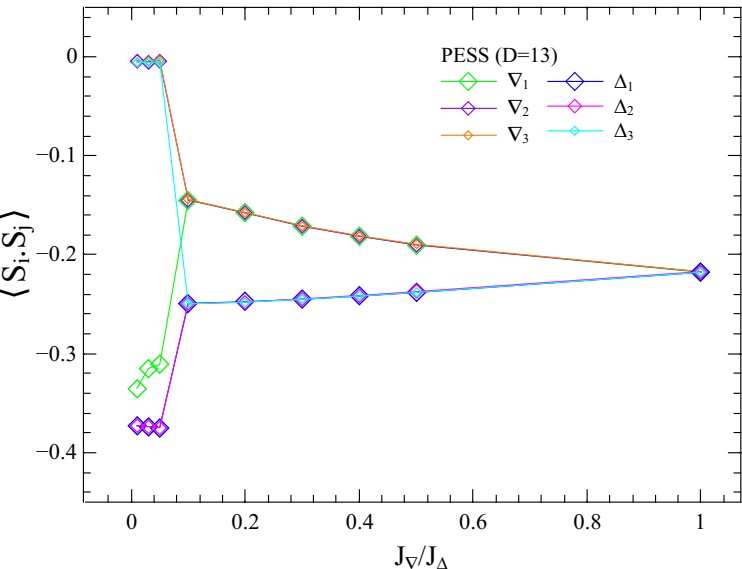

Figure 11: (Color online) Nearest-neighbor spin-spin correlations (obtained with PESS ($D = 13$)) on the three links of an upward triangle, $\triangle_1, \triangle_2, \triangle_3$ and on the links of a downward triangle $\triangledown_1, \triangledown_2, \triangledown_3$ (see also Fig. 6) which confirms the preservation of $C_3$ rotational symmetry at the level of each individual triangle in the U(1) QSL phase and breaking of the $C_3$ symmetry in the nematic phase. The values for $\triangle_1, \triangle_2$ are the average value of the correlation on the two links.

cylinder. However it has not been possible to extrapolate the results to infinite $D$, and strictly speaking one cannot exclude that the U(1) spin liquid remains stable below $J_{\triangledown}/J_{\triangle} \simeq 0.05$ on the basis of the present results.

In view of the competing phases that have been proposed over the years for the strong breathing limit, let us conclude with a critical review of the situation. The early mean-field theory of the effective model [14] had identified the short-range RVB basis as a promising variational basis, a conclusion partly confirmed in Ref. [41]. Interestingly, recent results based on VMC [42] favour a VBC configuration that is consistent with a particular member of the short-range RVB basis in which all pairs of triangles coupled by a singlet bond are parallel to a given direction. Such a state breaks the rotational symmetry in real space, but it is different from the nematic state reported here because it breaks an additional mirror symmetry.

The results of the present paper and of Ref. [44] on one hand, and the results of a very recent investigation based on an improved simplex RVB PEPS ansatz on the other hand [59], all suggest that energy can be further gained with respect to this VBC state by additional quantum fluctuations, but in different ways.

In the lattice-nematic state, these fluctuations take place along the chains of the phase with broken rotational symmetry. The additional mirror (or equivalently unit lattice translation) that is broken in the VBC state is restored, but the rotational symmetry is not. The ground state consists of very weakly coupled chains. On each chain, the wave-function is typical of a spin-1/2 critical chain, with correlations that require to go beyond the short-range RVB basis and to include longer-range singlets. So the energy gain w.r.t. the VBC has to do with the development of long-range correlations.

By contrast, in the RVB phase, the full symmetry is restored, the energy gain comes from resonances between short-range RVB configurations, and the spectrum is gapped.

Quite interestingly, the energies of these two states are very close, and even if that of the RVB spin liquid is higher for $D = 12$[1], it is not possible to make a reliable prediction for what happens in the infinite $D$ limit. So, if it seems very likely that there is a phase transition at strong breathing anisotropy, the actual nature of the ground state in that limit, a critical lattice-nematic state or a gapped RVB state, should still be considered as an open question. This is a motivation to further improve tensor-network algorithms, and work is in progress along these lines.

In any case, the conclusion that the transition takes place at a very small value of $J_{\bigtriangledown}/J_{\triangle}$, of the order of 0.05, is in agreement with the recent experiments on vanadium compounds [37–39] in which evidence for a gapless U(1) ground-state was observed on a kagome lattice with breathing anisotropy at $J_{\bigtriangledown}/J_{\triangle} \approx 0.55$.

Finally, we would like to note that the TN algorithm that we have developed for the kagome lattice can also be used for further investigations of the BKH model in the presence of a magnetic field to capture possible magnetization plateaus and their underlying exotic phases. This is left for future investigation.

## Acknowledgements

S.S.J. acknowledges FM for hospitality during his stay at EPFL. Tensor network simulations were performed on the FIDIS HPC cluster at EPFL and the ATLAS HPC cluster at DIPC. DP acknowledges support by the TNSTRONG ANR-16-CE30-0025 and TNTOP ANR-18-CE30-0026-01 grants awarded by the French Research Council. FM acknowledges the support of the Swiss National Science Foundation. Fruitful discussions with F. Pollmann, A. Kshetrimayum, Hong-Hao Tu and P. Schmoll are also acknowledged. We also thank C. Repellin for providing details about the DMRG data of Ref. [44] in private communications [58].

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
