# Peer review of "Spin-$\frac{1}{2}$ kagome Heisenberg antiferromagnet with strong breathing anisotropy"

_SciPost Physics_

## Round 1 · Referee Report · Anonymous (Referee 1) · 2020-1-31

Strengths

Provide comprehensive comparison with previous studies.

Weaknesses

The dependence on the bond dimension D is not carefully discussed. Due to which it is unclear about the validity of some claims, e.g. the value of critical anisotropy $J_\triangle/J_\triangledown$, the nature of the phase transition.

Report

This paper studies the phase diagram of the spin-1/2 Heisenberg model with breathing anisotropy. The same model has been previously studied by using the DMRG and VMC, the current paper uses a different numerical method, i.e. iPEPS. The work is generally consistent with previous study, namely the spin liquid phase in the isotropic limit survives under strong breathing anisotropy. For the extreme breathing limit, this work finds a gapless phase with spatial nematic order (i.e. a state with spontaneously lattice rotation breaking). This result is consistent with the previous DMRG study but different from the VMC calculation. Moreover, this work finds the critical anisotropy to be around 0.05.

Requested changes

  1. On page 6 and figure 5, the authors find that the scaling of the ground-state energy with respect to bond dimension follows a power law. Therefore, they claim a gapless state. I do not find this convincing. Is there any theoretical proof that a power law scaling with respect to bond dimension would necessarily imply a gapless state? And why does it indicate a U(1) (rather than Z2) gapless spin liquid? The author shall provide more concrete discussion or clarify that it is an empirically numerical expectation.

A minor issue is, based on the plot it is hard to tell the scaling behaviour is a power law. Can the authors contrast it with other scalings such as $e^{D/a}$ and $\log D$?

  1. In Figure 7, the correlation function at large distance $r$ clearly falls off the power law decaying. It is likely to be an artifact of finite bond dimension $D$. It would be good to plot the $D$-dependence of correlation function to show the trend that the power scaling becomes better as $D$ increases.

  2. The paper claims a first order phase transitions at $J_\triangle/J_\triangledown=0.05$ based on the calculation with $D=13$. Is this statement sensitive to $D$? More concretely, i) does the critical value shift as $D$ is changing? ii) does the phase transition becomes more continuous as $D$ increases? It is possible that the observation of a first order phase transition is an artifact of finite $D$.

  • validity: -
  • significance: -
  • originality: -
  • clarity: -
  • formatting: -
  • grammar: -

Author:  Saeed Jahromi  on 2020-09-23  [id 1017]

(in reply to Report 1 on 2020-01-31)

See attached file

Attachment:

Reply-Reviewer-1.pdf

---

## Round 1 · Referee Report · Anonymous (Referee 2) · 2020-2-25

Strengths

  1. The authors use two numerical methods to study spin-1/2 Heisenberg model on extended Kagome lattice (with breathing anisotropy). This paper contributes to a recent DMRG study of the same model by F. Pollmann et.al., especially in the limit of large breathing anisotropy.

  2. The paper presents an extensive set of state-of-the-art results of iPEPS and PESS calculations, and compare with the results of calculations obtained with other techniques.

  3. The paper confirms the phase transition between a spin-liquid and a lattice-nematic state for large breathing anisotropy, found by F. Pollmann et.al.

Weaknesses

  1. The value of anisotropy separating the spin-liquid and the lattice-nematic state differs (rather significantly) from that of F. Pollmann. The authors mention in the conclusion that this difference is due to finite size effects of the previous study. However, this point does not seem to be sufficiently justified.

Report

The paper contributes to the studies of the Heisenberg model on a Kagome lattice with new results obtained using iPEPS and PESS methods, and it will be of interest to frustrated magnetism community.

The model with breathing anisotropy is believed to capture the properties of new spin-liquid candidate material (vanadium oxyfluoride), so the study is also relevant to current experiments.

The numerical methods developed in the paper could be used to study the model in the presence of magnetic field, which is an interesting problem.

Extensive numerical results presented in the paper will also serve as a useful benchmark for further numerical studies.

Therefore I recommend this paper for publication in SciPost.

Requested changes

  1. The authors suggest that the quantum phase transition is the first-order. This statement does not seem to be sufficiently justified by the results presented in Fig. 8. Perhaps add further confirmation of the statement.

  2. In Fig. 3 it would be useful to show the ground state energy obtained using other methods, similar to Fig. 4. E.g. show the energies obtained using DMRG by F. Pollmann et. al., and Yau et. al.

  3. In Fig. 3. there is minus sign missing in the power laws of the fits shown in the inset.

  4. Introduce acronyms SU and FU early (on page 4), and clarify the meaning. One can also remove CDMRG word from the figures for clarity.

  5. In most figures the results of Ref. [44] are shown as DMRG, but in Fig. 5 as Nematic-DMRG. Perhaps make this uniform throughout, and make a reference to [44] in the text for the figures.

  6. Show results for the isotropic case in Table 1. Perhaps, make in bold the lowest energy value in each row.

  7. In Ref. [44] the critical value of breathing anisotropy was found to be 0.14. It would be good to cite this value in the beginning of section 4 of the paper, and compare with the value of 0.05 found in this paper.

  8. Perhaps add a more extended discussion of the reason why there is a discrepancy between the critical point found in this paper and Ref. [44]. The current explanation does not seem to be justified.

  9. If possible, show scaling of the gap for different values of anisotropy.

  • validity: high
  • significance: high
  • originality: good
  • clarity: good
  • formatting: excellent
  • grammar: good

Author:  Saeed Jahromi  on 2020-09-23  [id 1018]

(in reply to Report 2 on 2020-02-25)

See attached file.

Attachment:

Reply-Reviewer-2.pdf

---

## Editorial Decision

resubmitted